# ABCA3 Deficiency—Variant-Specific Response to Hydroxychloroquine

**DOI:** 10.3390/ijms24098179

**Published:** 2023-05-03

**Authors:** Xiaohua Yang, Maria Forstner, Christina K. Rapp, Ina Rothenaigner, Yang Li, Kamyar Hadian, Matthias Griese

**Affiliations:** 1Dr. von Haunersches Kinderspital, German Center for Lung Research, University of Munich, Lindwurmstr. 4a, 80337 Munich, Germany; xiaohua.yang@med.uni-muenchen.de (X.Y.); maria_elisabeth.forstner@med.uni-muenchen.de (M.F.); christina.rapp@med.uni-muenchen.de (C.K.R.); liyanggzyx@outlook.com (Y.L.); 2Research Unit Signaling and Translation, Helmholtz Zentrum München, Ingolstädter Landstr. 1, 85764 Neuherberg, Germany; ina.rothenaigner@helmholtz-munich.de (I.R.); kamyar.hadian@helmholtz-munich.de (K.H.); 3Medical College, Chongqing University, Chongqing 400044, China

**Keywords:** ATP-binding cassette subfamily A member 3, ABCA3, interstitial lung disease, ILD, hydroxychloroquine, HCQ, diffuse parenchymal lung disease, chILD, childhood interstitial lung disease

## Abstract

Biallelic variants in *ABCA3*, the gene encoding the lipid transporter ATP-binding cassette subfamily A member 3 (ABCA3) that is predominantly expressed in alveolar type II cells, may cause interstitial lung diseases in children (chILD) and adults. Currently, there is no proven therapy, but, frequently, hydroxychloroquine (HCQ) is used empirically. We hypothesized that the in vitro responsiveness to HCQ might correlate to patients’ clinical outcomes from receiving HCQ therapy. The clinical data of the subjects with chILD due to ABCA3 deficiency and treated with HCQ were retrieved from the literature and the Kids Lung Register data base. The in vitro experiments were conducted on wild type (WT) and 16 mutant ABCA3-HA-transfected A549 cells. The responses of the functional read out were assessed as the extent of deviation from the untreated WT. With HCQ treatment, 19 patients had improved or unchanged respiratory conditions, and 20 had respiratory deteriorations, 5 of whom transiently improved then deteriorated. The in vitro ABCA3 functional assays identified two variants with complete response, five with partial response, and nine with no response to HCQ. The variant-specific HCQ effects in vivo closely correlated to the in vitro data. An ABCA3^+^ vesicle volume above 60% of the WT volume was linked to responsiveness to HCQ; the HCQ treatment response was concentration dependent and differed for variants in vitro. We generated evidence for an *ABCA3* variant-dependent impact of the HCQ in vitro. This may also apply for HCQ treatment in vivo, as supported by the retrospective and uncontrolled data from the treatment of chILD due to ABCA3 deficiency.

## 1. Introduction

The ATP-binding cassette subfamily A member 3 (ABCA3), which is a lipid transporter expressed in alveolar type II cells, is localized at the limiting membrane of the lysosome-related lamellar body (LB) and involved in surfactant metabolism, which is crucial for normal breathing [1,2]. The pathogenic variants in *ABCA3* are the most frequent monogenetic cause for interstitial lung disease in children (chILD) associated with surfactant dysfunction [3,4]. Nonsense and frameshift variants in *ABCA3* commonly lead to a “null” phenotype with neonatal respiratory distress syndrome and death within the first months of life. In-frame insertions/deletions or missense *ABCA3* variants may result in “hypomorphic” variants, which may be compatible with survival and associated with chronic interstitial lung disease (ILD) in children and adults [5,6,7].

Currently, no specific therapy exists for lung diseases related to pathogenic *ABCA3* variants. Hydroxychloroquine (HCQ) is an empirical treatment for chILD, either as a single agent or concomitant with other medicines [8]. The results of a recent randomized controlled phase 2 trial of HCQ in 35 patients with various forms of fibrosing chILD conditions, including 5 patients with ABCA3 deficiency, did not identify an overall HCQ treatment effect [9]. The response to HCQ from the reported patients with disease-causing *ABCA3* variants varies from no response to partial response with chronic condition to complete recovery [10,11,12]. As ABCA3-variant-specific response to HCQ treatment from randomized controlled trials are to our best knowledge not expected soon, it is worthwhile to retrospectively compile HCQ treatment responses from such patients and to analyze their responses in relation to detailed in vitro assessment of such *ABCA3* variants.

HCQ is a weak base and has lysosomotropic activity [13,14,15]. On the cellular level, ABCA3 is localized in the limiting membrane of LBs that are derived from the lysosomal compartment [16,17]. The impact of HCQ on the intracellular localization of ABCA3, its processing, the LB volume, and lipid transport has not been extensively studied in appropriate cellular models. 

We hypothesized that, in case there was a direct correlation of the ABCA3-variant-specific in vitro and in vivo response to HCQ treatment, the in vitro response could predict the observed clinical effect in patients carrying such variants. Therefore, we systematically analyzed the clinical outcomes of patients with disease-causing *ABCA3* variants treated with HCQ, explored the effect of HCQ on a large body of ABCA3 variants at the cellular level, and evaluated the correlation of HCQ responses in vitro and in vivo.

## 2. Results

### 2.1. Outcome of Patients with ABCA3-Related Lung Disease and HCQ Therapy

We identified 48 patients with ABCA3-related lung disease from the Kids Lung Register, of which 30 were treated with HCQ and had sufficient clinical records. In addition, 9 HCQ-treated patients with ABCA3 deficiency were identified in the literature [11,18,19,20,21,22,23,24]. Thus, 39 patients were assessed in detail (Figure 1). On average, 46.1% were female and 84.6% (33/39) developed respiratory symptoms within the neonatal period. The median age at which HCQ treatment started was 0.3 (0.1–15.8) years, at an initial assessment of HCQ therapy at 0.8 (0.1–17.1) years and at a last assessment at 1.7 (0.3–13.0) years (Appendix A). 

At the initial assessment, 15 of 39 patients’ respiratory condition deteriorated with HCQ therapy. The other 24 patients had respiratory improvement with HCQ therapy and 19 of them received their last assessment. Of note, five patients had a transient response, but deteriorated at the last assessment despite receiving continued HCQ therapy. Finally, 20 patients had respiratory deterioration, while 19 patients had improved or unchanged respiratory condition with HCQ treatment (Figure 1).

The display of the individual patient data demonstrated that the start and duration of the HCQ treatment varied largely (Figure 2). The patients with respiratory improvement were older than those with respiratory deterioration (*p* value 0.0554), as was the age at their initial and last assessment of the HCQ therapy response (Appendix A). Most of the patients (86.4%) who survived during the study period responded to HCQ treatment (Figure 2).

Of the 20 patients lacking a definite respiratory improvement, 15 died and 2 received lung transplants (Figure 2). Of note, all three patients (patient 21, 25, and 33) with null/null variants died before the first year of age despite receiving HCQ therapy (Appendix A). Of the ten patients with null/hypomorphic variants, six had respiratory deterioration and four improved (Appendix A). 

### 2.2. Variant-Specific and Dose-Dependent Effects of HCQ on ABCA3 In Vitro Using a High-Content Screening (HCS) Assay

To explore the variant-specific effects of HCQ on ABCA3-deficient cells, we probed HCQ treatment in vitro in 16 different missense *ABCA3* variants. The 16 variants were selected based on their frequent identification (E292V), their homozygote appearances (such as V1399M, P248S, and Q1045R), and the varying outcomes (such as A1046E, G1421R, D953H, and E1364K) observed in patients with ABCA3-related ILD. M760R and Q215K were selected to add evidence to the in vitro response of ABCA3 variants without a detectable ABAC3^+^ vesicle (G571R) to HCQ (Figure 3, Appendix A). The HCS method was used to test different HCQ concentrations in the selected variants (overview in Appendix A).

The read out in the HCS assay was the percentage of WT-like cells with their characteristic LB-like organelles (% WT-like cells) (Figure 3). In WT cells, the mean and range of the nSD were 11.4% (range 9.31–13.90%). Administering 10 μM HCQ increased the % WT-like cells by 137.4%, whereas the other concentrations had no effect. All ABCA3 variants started in the “defective” range (beyond mean −3 nSD). With an increasing HCQ concentration, Q1045R and E292V moved into the “normal” range (mean −1 nSD) (Figure 3), and G202R, F1077I, E1364K, and D953H moved into the “impaired” range (mean >−1 and <−3 nSD), thus indicating their responsiveness to HCQ. All these effects were HCQ-dose-dependent and statistically significant (Appendix A). Of note, the largest increase in G202R was achieved with 30 µM HCQ treatment; a higher dosage did not further improve the % WT-like cells. The other variants remained “defective % WT-like cells” with HCQ treatment (Figure 3), including G1421R, V1399M, G1314E, and A1046E, which showed minor but statistically significant increases at different specific HCQ doses (Figure 3, Appendix A).

### 2.3. Detailed Evaluation and Validation of HCQ Response in Small-Format Assays (In Vitro)

To confirm the variant-specific effects of HCQ, we explored the effect of 10 µM HCQ on WT and 9 of the 16 variants in validated small-format assays (Appendix A). 

We assessed the processing of ABCA3 by determining its proteolytic cleavage using Western blotting. The HCQ treatment increased the ratio of 170/190 kDa in the variants G1421R, V1399M, A1046E, D953H, Q1045R, and E292V to a normal level, thus indicating that HCQ enhanced the full processing of these mutant ABCA3 proteins (Figure 4A). The HCQ treatment did not change the ratio of 170/190 kDa in the variants M760R, Q215K, and E1364K (Figure 4A and Appendix A).

The volume of ABCA3^+^ vesicles and transport of propargyl-choline into the ABCA3^+^ vesicles indicated the pumping function of ABCA3, assessed using immunofluorescent staining (Figure 4B,C and Appendix A). In the variants A1046E, E1364K, D953H, Q1045R, and E292V, both the vesicle volume and the amount of propargyl-choline in the ABCA3^+^ vesicle increased to normal levels upon HCQ treatment. No effect was seen in the variants M760R, Q215K, G1421R, and V1399M (Figure 4B,C). 

We also compared the results from HCS (Figure 4D) directly with the data from the Western blotting and immunofluorescent staining (Figure 4A–C). The consistency of these results validated the utility of the HCS approach to assess the HCQ treatment effects in vitro.

### 2.4. HCQ Acts on the Lysosome-Related Components in ABCA3 Transfected Cells

We assessed the subcellular distribution of ABCA3 variants E292V, Q1045R, and D953H with and without HCQ treatment. Those variants were responsive to HCQ in four in vitro assays and displayed a vesicle-like distribution, mainly co-localizing with the lysosomal marker CD63 and having no co-localization with the ER marker calnexin (Figure 5 and Appendix A). This indicated that HCQ had effects on lysosome-related ABCA3^+^ vesicles.

### 2.5. Comparison of In Vivo and In Vitro Results

The individual patients’ responses, their genetic variants, co-medication, and respiratory outcomes are listed in Appendix A. The semi-quantitative sum responses to HCQ in vitro of the two ABCA3 variants were correlated to the respiratory outcomes of patients with ABCA3-related lung disease (Spearman’s r value 0.54. Figure 6A). There was a positive correlation between the volume of the ABCA3^+^ vesicle and average response to HCQ in vitro in the ABCA3 variants (Spearman’s r value 0.84. Figure 6C). The volume of the mutated ABCA3^+^ vesicle over 60% of that of the untreated WT ABCA3^+^ vesicles may predict the responsiveness to HCQ treatment in vitro.

Of the 35 subjects included in the HCQ trial, 5 had ABCA3 deficiency [9]. The variants (R208W/c.3863-98C>T and R280C/A96D) in two of these patients were not investigated in vitro. For all other variants, the in vitro tests were completed in agreement with the double-blinded in vivo observations. The patients carrying E292V/G571R responded to HCQ introduction and withdrawal according to the protocol criteria. The patient carrying D953H/F1077I responded to HCQ introduction, and withdrawal was not performed. In the patient with P248S/P248S, the double-blindly assessed HCQ withdrawal did not induce deterioration, and the HCQ introduction was not performed. This was patient 2 in this study, whose respiratory condition improved with HCQ concomitant with systemic steroids and azithromycin in the initial assessment but remained unchanged in the last assessment (Figure 2, Appendix A). 

Another patient with discordant in vitro/in vivo results was patient 22 from the retrospective case collection (carrying G1421R/Q1045R), who had respiratory deterioration with HCQ treatment, whereas the sum response to HCQ in vitro of these two variants was positive (Q1045 was responsive (+2) and G1421R was not responsive (+0.5) to HCQ in vitro) (Figure 6, Appendix A). When re-calculating without patients 2 and 22, the in vitro/in vivo correlation improved (Spearman’s r value 0.89. Figure 6B).

## 3. Discussion

For chILD caused by the biallelic pathogenic *ABCA3* variants, there are no proven treatments available [3,10]. Based on the empirical data, HCQ is frequently used [9]. Here, we detailed the in vivo treatment response to HCQ in 39 children carrying specific *ABCA3* variants and correlated this to in vitro assessments of HCQ treatment. It was shown that the clinical responses of children to HCQ strongly depended on the *ABCA3* variants they carried, and that the in vitro and in vivo responses to HCQ treatment were closely correlated. The more the ABCA3^+^ vesicles of variants morphologically resembled those of WT, the better their response to HCQ in vitro was. A concentration dependence of the maximal HCQ effect in different variants in vitro was also demonstrated. 

The type and associated loss of function of the *ABCA3* variants strongly predicts the outcome [5,6,7,25]. This pattern was not significantly altered by administering HCQ, i.e., HCQ therapy could not protect patients with biallelic loss of function variants (“null/null” variants) from early death or lung transplant within the first year of life. Currently, the clinical course of patients carrying “null/hypomorphic” or “hypomorphic/hypomorphic” variants is difficult to predict. Facing an uncertain natural course of recovery after the initial manifestation of ABCA3 deficiency, the additional impact of the treatments may only be objectified in double-blind controlled trials. The treatment response determined from the retrospective analysis as performed here for most of the variants may contribute important though weaker evidence for HCQ effects, particularly when together with a good correlation to the variant-specific in vitro responses. No in vivo/in vitro correlation would imply erroneous in vitro responses or, more likely, due to uncontrolled retrospective evaluation, incorrect in vivo observations.

As long-term HCQ treatment may potentially cause severe side effects, it is desirable to generate the best possible treatment predictions [11]. In vitro evidence may be helpful as successfully demonstrated for cystic fibrosis and other diseases [26,27]. Our data on the impact of HCQ on many ABCA3 variants at the cellular level identified an overall close in vitro–in vivo correlation for HCQ treatment. Of interest were potential explanations for two unexpected deviations from the correlation. The positive evaluation of the HCQ effect for the ABCA3 variant P248S in the retrospective study was likely biased because of the co-treatment. Thus, we put more weight on the result of the double-blindly determined treatment effects, although these may be false negatives. Even though the sum response to HCQ in vitro of G1421R and Q1045R was positive, patient 22 had an early fatal respiratory postnatal course. This might be due to the insufficient correction of the ABCA3 function or additional complications, before the HCQ therapy could fully take effect.

The higher the volume of the ABCA3^+^ vesicles of the probed variants, the more likely a response to HCQ in vitro was. A vesicle volume of no less than 40% of the WT volume appeared to be a threshold for HCQ responsiveness. In the variants A1046E, E1364K, D953H, Q1045R, and E292V, 10 µM of HCQ increased both the volume of the ABCA3^+^ vesicles and the transport of propargyl-choline into ABCA3^+^ vesicles to normal levels, thus indicating that HCQ might potentiate the lipid pumping activity of ABCA3 [28]. In addition, the intracellular process of proteolytic cleavage was enhanced, supporting regulatory feedback on the promotion on ABCA3 synthesis [16,29].

Conversely, in variants such as M706R, Q215K, G1421R, and V1399M, HCQ had little or even no effect. This may be explained by the observation that these variants lack any detectable lysosome-related ABCA3^+^ vesicles, which may be essential components for HCQ to act on [16,26,30]. Due to HCQ’s lysosomotropic activity as a weak diprotic base, HCQ accumulates within lysosomes and other intracellular acidic compartments, and possibly acts by increasing the pH of those intracellular compartments [14,15]. There is evidence that the proteolytic cleavage of ABCA3 takes place in lysosome-like compartments [16,17]. Our immunofluorescent imaging confirmed the subcellular co-localization of variants E292V, Q1045R, and D953H with the lysosomal marker CD63. Exploring the effect of HCQ on lysosomal enzymes’ activities may assist to analyze ABCA3 processing further [26].

With the increasing HCQ concentration, most of those ABCA3 variants with detectable ABCA3^+^ vesicles responded to HCQ treatment in vitro. Of interest, the largest increase in % WT-like ABCA3 in G202R was achieved with 30 µM HCQ treatment but not with a higher dosage. This supports the assumption that different doses for the maximal effect in different variants may also play a role in vivo. A randomized controlled phase 2 trial of HCQ in 35 patients with chILD showed considerable inter-individual variance of HCQ whole blood levels [9]. Future studies of chILD caused by ABCA3 deficiency should assess pharmacokinetics and dose efficacy. An open-label prospective pilot study reported that whole blood concentrations above 750 ng/mL HCQ were associated with a significant improvement in refractory cutaneous lupus erythematosus in adults [31]. Therefore, we suggest measuring HCQ blood levels in children even if prescribed doses adapted to the patient’s weight.

The limitations of this study include that the clinical data were analyzed retrospectively. Differentiating if the improvement in patients under HCQ treatment was in part or solely due to this treatment or if it occurred independently with time was not possible due to the uncontrolled observational study design. In this line, the additional treatments the patients received as well as other endogenous (i.e., additional genetic variants in other genes [32]) or extrinsic factors may have affected the observed responses to HCQ treatment. Only data from prospective studies will be able to unequivocally determine the treatment effects. For the in vitro experiments, A549 cells were used as a well-established model of *ABCA3* variant functional assessments [33,34]; however, in this model, only the homozygous conditions of the variants can be expressed. To rate the compound heterozygous constellation, we used the sum of the responses of the two variants. This appears to be an appropriate extrapolation from other ABC transporters, without considering the potential interactions of the variants [32,35]. In the future, patient-derived induced pluripotent stem cells or cells engineered to carry *ABCA3* variants may provide an opportunity to explore the role of individual patient-specific genetic backgrounds. Important environmental influences in the clinical course of the patients likely play a key role and were not assessed in this study [36].

## 4. Materials and Methods

### 4.1. Response of Patients with ABCA3 Deficiency to HCQ (In Vivo)

The patients identified in the program for rare lung diseases of the Kids Lung Register (KLR) between 1 January 2001 and 1 July 2020 with homozygous or compound heterozygous *ABCA3* variants and treated with HCQ were included in this study, as well as cases retrieved from PubMed literature research restricted to articles in English and published before 1 July 2020 (Figure 1). The search strategy included the terms “ABCA3, lung disease, treatment or therapy, hydroxychloroquine”. In patient 12, although one *ABCA3* variant was identified on only one allele, the similarity of her clinical presentation and the severe nature of the lung disease suggested that this patient probably had a second pathogenic *ABCA3* variant on the other allele, which was not detected using our current sequencing strategy, such as mutations within introns or regulatory regions or large rearrangements. Therefore, this case was included in this study to present as much clinical information as possible but was not included for correlation with the HCQ in vitro responses. Of note, in patient 3 and patient 39, there were more than one *ABCA3* variant on the other alleles (Appendix A).

The clinical data were collected manually (see variable list in Appendix A) from birth to the end of the follow up, focusing on respiratory diseases and their treatments. With advances in rapid diagnostic testing for surfactant dysfunction disorders, most of the diagnoses were made genetically and not using histopathology. The ABCA3-deficient lung histology is not specific to the light microscopic level, but electron microscopy may demonstrate characteristic abnormal lamellar bodies [7]. All observations on lung disease onset and HCQ treatment were arranged according to the patients’ ages (Figure 2). The key time points were the age at the lung disease onset and the age at the start of the HCQ therapy. The latter was defined as the age when the patient received HCQ for the first time. If a patient had received HCQ before but there was no information for that period, then that period was not considered in this study. We identified the corresponding ages when the clinical response to HCQ therapy was assessed for the first time (initial assessment) and when the latest assessment under HCQ therapy was performed (last assessment). The criteria were listed in Appendix A. A respiratory improvement was scored as 1, no change as 0, and a deterioration as −1. Then, the average score of the respiratory outcome was calculated by dividing the sum of the initial and last assessments by the number of assessments (Appendix A).

In addition, the results of the patients with ABCA3 deficiency that were included in a recently published clinically double-blind randomized controlled trial were extracted and compared to the in vitro results of the assessed variants [9].

### 4.2. Response of Cells Expressing ABCA3 Variants to HCQ (In Vitro)

#### 4.2.1. Cell Culture, Viability, Western Blotting, and Immunofluorescence Staining

The A549 cells stably transfected with ABCA3 and expressing HA-tagged wild type (WT) or mutant ABCA3 protein (ABCA3-HA) were generated and cultured as previously described [30,37]. The cells were treated with different concentrations of HCQ in phenol red free RPMI medium/10% fetal bovine serum. The viability was quantified by measuring the specific cleavage of yellow XTT tetrazolium salt (Sigma Aldrich, Taufkirchen, Germany) to orange formazan in the presence of phenazine methosulfate (Sigma Aldrich, Taufkirchen, Germany) at 450 nm in a spectrophotometer at from 5 to 40 µM for up to 72 h (Appendix A). The immunoblotting with 15 µg protein per lane, immunofluorescent staining, and quantification were performed as previously described [28,30,34].

#### 4.2.2. High-Content Screening Method (HCS)

The screening procedure and automated image analysis were applied as previously described [38]. Briefly, the cell culture and treatment and the immunofluorescence staining were performed manually. The multiparametric image analysis was performed using Columbus software version 2.8.0 (PerkinElmer, MA, US). We conducted a phenotypic cell-based assay to autonomously identify the ABCA3 WT-like or mutant-like cells by using machine-learning algorithms. After training, the software identified the linear combination of the most relevant properties that determined an effective discriminator for “WT-like cells” and “mutation-like cells”.

#### 4.2.3. Expression of Results of the Functional Assays, Criteria for In Vitro Response to HCQ, and Statistical Analysis

For convenient and inter-laboratory comparisons, the results of all the functional assays, i.e., % WT-like cells in the HCS assay (Figure 3 and Figure 4D), proteolytic cleavage of ABCA3, volume of ABCA3^+^ vesicle, and transport of propargyl-choline into ABCA3^+^ vesicle (Figure 4), were expressed as a percentage of the WT condition (%WT nt). The “normal” ranges were defined based on the variations observed in the experiments with ABCA3 WT cells. For each assessed parameter, the standard variation (SD) of a WT experiment was divided by its mean (normalized SD (nSD)). Then, all the nSD were used to calculate their mean and range. We categorized the extent of deviation from “normal”, if the results in the mutants (%WT) fell within the range of −1 nSD, as “impaired” within −3 nSD and as “defective” if beyond −3 nSD.

The responses to HCQ were scored as follows: If the HCQ treatment maximally improved the readout variable into the impaired range, this was called a partial response to HCQ and scored as +1. If the HCQ treatment improved the readout variable into the normal range, or stayed within normal range, this was called a complete response to HCQ and scored as +2. No increase outside the defective range was defined as no response to HCQ and scored as 0. The average score of the response to HCQ in vitro was defined as the mean response from different assays performed for a variant (Appendix A). The responses to HCQ were solely judged on these criteria and not on statistical comparisons to untreated cells (see below).

For the correlation of the HCQ in vitro responses to those in patients, we calculated the sum score of the variants present in a patient (Appendix A). The sum response in vitro of the two *ABCA3* variants ≥2 was defined as responsive to HCQ treatment in vitro. In case one variant was not explored, this variant pair was excluded from correlation analysis to the in vivo data. 

According to the American College of Medical Genetics and Genomics (ACMG) guidelines [39], the interpretations of the *ABCA3* variants were classified as “benign”, “likely benign”, “uncertain significance”, “likely pathogenic”, and “pathogenic” and scored as 1, 2, 3, 4, and 5, respectively. If there was more than one variant identified on one allele of a patient, the scoring was based on the most severe variant.

A one-way ANOVA with a Dunnet’s post hoc test was used to compare the mean values among multiple treatment groups, i.e., to explore the effect of different HCQ concentrations in comparison to no treatment in WT or variants (Figure 3). *t*-Tests were used to compare the mean value between the two groups (Figure 4). A Spearman correlation analysis was used to identify the correlation of the HCQ in vitro responses to those in the patients. *p* values < 0.05 were considered as statistically significant (GraphPad Prism 7.0, La Jolla, CA, USA).

## 5. Conclusions

We generated evidence for the *ABCA3*-variant-dependent impact of HCQ in vitro. This may also apply for HCQ treatment in vivo, as supported by the retrospective and uncontrolled data from the treatment of chILD due to ABCA3 deficiency (Figure 7). In the future, randomized controlled and prospective studies in children with ABCA3-related lung disease, in vivo drug level monitoring, and/or other personalized treatments based on advanced cellular models are expected.

## Figures and Tables

**Figure 1 ijms-24-08179-f001:**
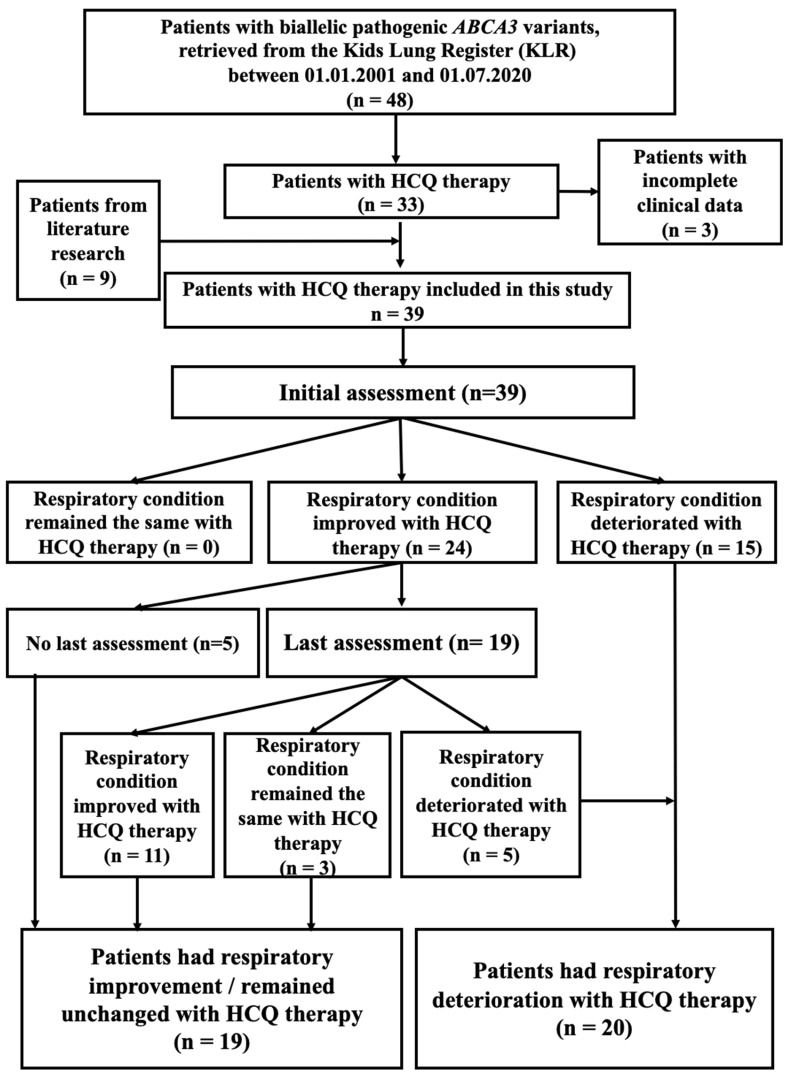
Flow chart of the cohort included in this study.

**Figure 2 ijms-24-08179-f002:**
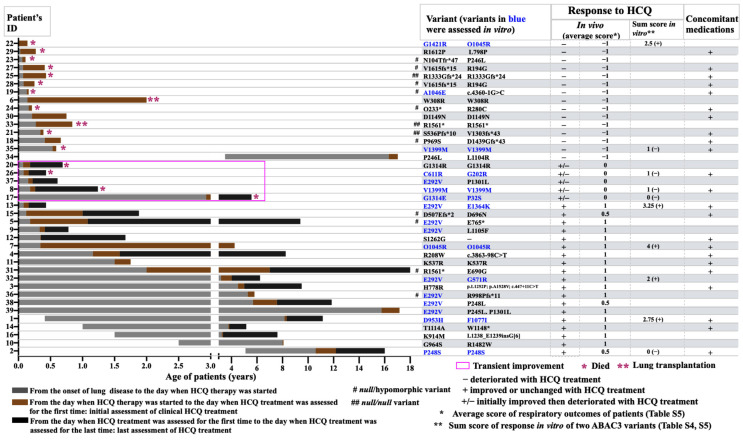
Cohort of 39 patients with pathogenic *ABCA3* variants and treated with HCQ, sorted by age of lung disease onset and clinical severity. Data of patients 3, 14, 15, 16, 27, 28, 29, 30, and 31 were retrieved from the literature, and the other cases were from the Kids Lung Register (see Appendix A). Start and duration of HCQ treatment varied largely. In most of the patients who survived a clinical response (in vivo, columns 3 and 4 from right) to HCQ treatment was noted, whereas, in those who dead, only transient or no responses were noted. The in vitro responses to HCQ (column 2 from right) where available were correlated to the in vivo responses.

**Figure 3 ijms-24-08179-f003:**
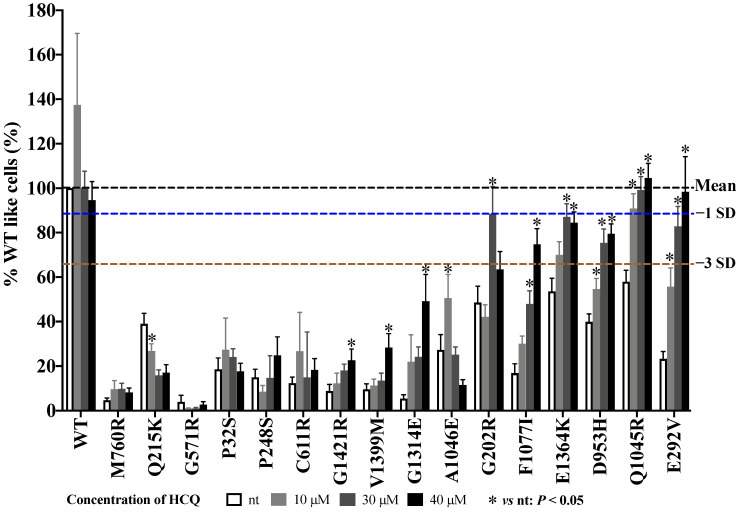
Response of 16 ABCA3 variants to HCQ in vitro assessed using high-content screening method. Results (% WT nt) were shown as means + S.E.M (Individual values were listed in Appendix A). Black dotted line indicated mean value. Blue dotted line indicated mean −1 nSD. Brown dotted line indicated mean −3 nSD. * indicates *p* value < 0.05.

**Figure 4 ijms-24-08179-f004:**
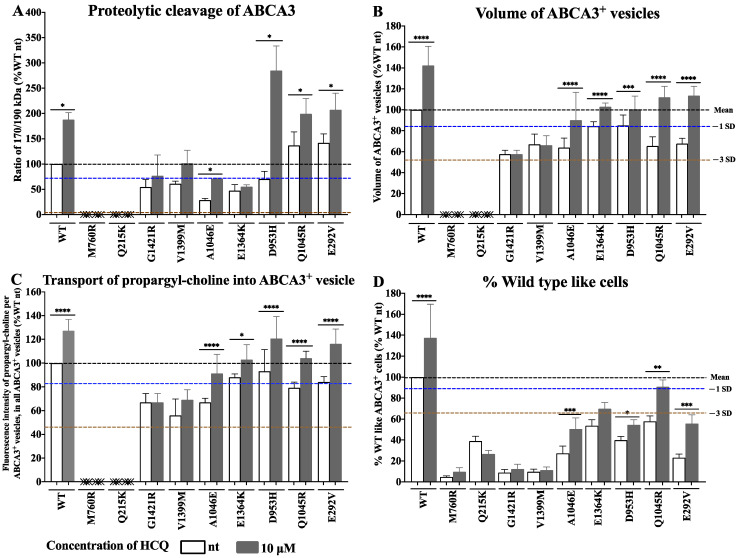
Nine ABCA3 variants assessed using three small-format assays for response to HCQ in vitro. Proteolytic cleavage of ABCA3 (**A**). Volume of ABCA3^+^ vesicles (**B**). Transport of propargyl-choline into ABCA3^+^ vesicles (**C**). Results from HCS with HCQ 10 μM treatment were shown as: % WT-like cells (**D**). Results (% WT nt) were shown as means + S.E.M of three independent experiments. * indicates *p* value 0.0332, ** indicates *p* value 0.0021, *** indicates *p* value 0.0002, and **** indicates *p* value < 0.0001. Black dotted line indicated mean value. Blue dotted line indicated mean −1 nSD. Brown dotted line indicated mean −3 nSD.

**Figure 5 ijms-24-08179-f005:**
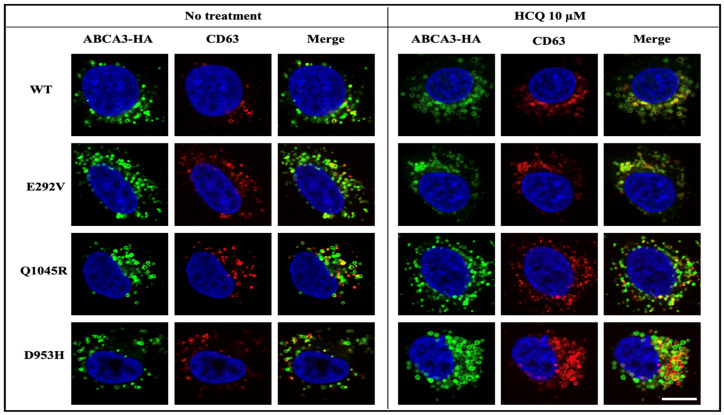
A549 cells stably expressing WT or mutated ABCA3-HA were treated with RPMI-1640 + 10% FBS (no treatment) or RPMI-1640 + 10% FBS-added HCQ 10 µM (HCQ 10 µM) for 24 h, and then stained for ABCA3-HA and lysosomal marker CD63. ABCA3-HA in green, CD63 in red, and DAPI in blue. Scale bar represents 20 μm.

**Figure 6 ijms-24-08179-f006:**
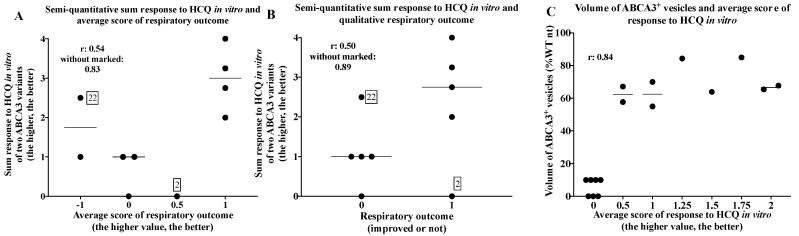
Correlation of semi-quantitative sum response to HCQ in vitro of two ABCA3 variants and average score of respiratory outcomes of patients (**A**), semi-quantitative sum responses to HCQ in vitro of two ABCA3 variants and qualitative respiratory outcomes of patients (**B**), and volume of ABCA3^+^ vesicles and average score of responses to HCQ in vitro by number of assays (**C**). Line indicates median value of individual values given in each column. Outlier were marked with patients’ ID in box (Appendix A).

**Figure 7 ijms-24-08179-f007:**
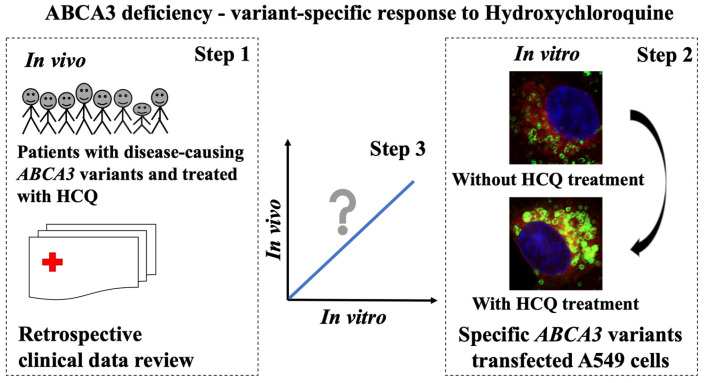
Summary of this study in pictogram: ABCA3 deficiency—variant-specific response to hydroxychloroquine. Step 1: systematically analyzed the clinical outcomes of patients with disease-causing *ABCA3* variants treated with HCQ. Step 2: explored the effect of HCQ on 16 ABCA3 variants on the cellular level. Confocal images show E292V-variant-transfected A549 cells without and with exposure to HCQ (10 μM). Step 3: evaluated the relation of the HCQ response in vivo and in vitro.

## Data Availability

Original data can be made available upon request.

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
