# Peer review of "ABCA3 Deficiency—Variant-Specific Response to Hydroxychloroquine"

_ijms, 2023, doi:10.3390/ijms24098179_

Round 1

Reviewer 1 Report

Please elaborate HCS (High-content screening method) in the section heading 2.2. as naive readers do not understand what is HCS assay.

Figure S3- please indicate HCQ beside  -/+ on the western images.

Figure S5- please indicate HCQ in the x-axis.

Figure S5- y-axis label is very hard to read, and can increase the font that is readable.

Reviewer 2 Report

The biallelic variants in ABCA3, predominantly expressed in alveolar type 2 cells have significant role in causing various interstitial lung diseases. Several reports are there for empirical usage of HCQ in ABCA3 deficiency. The author’s hypothesis for a direct correlation of ABCA3 variant response to HCQ is quite interesting. The systematic analysis involving clinical outcomes of patients with disease-causing ABCA3 variants in response to HCQ at cellular level and correlation of HCQ response is promising.

Major Comments:

1.    Please provide the histopathology of ABCA3-deficient lung tissue demonstrating interstitial thickening and septal fibrosis related to your study.

2.    Heterozygosity for ABCA3 mutations have been reported for modifying the SFTPC mutation, please shed some light on this context in your study.

3.    It will be a good idea to measure β-Glucosidase levels as a measure of activity of lysosomal enzyme in ABCA3 transfected cells in response to HCQ.

Minor Comments:

1.    Please include the hypothesis and important findings as a summary in pictorial figure format for easy understanding.

2.    In Figure 2, please include the results obtained briefly in the figure legend.
